# Ultrasensitive Electrochemiluminescence Immunoassay Based on Signal Amplification of 0D Au—2D WS_2_ Nano-Hybrid Materials

**DOI:** 10.3390/bios13010058

**Published:** 2022-12-30

**Authors:** Qile Li, Ke Xu, Haipeng Zhang, Zengguang Huang, Chao Xu, Zhen Zhou, Huaping Peng, Linxing Shi

**Affiliations:** 1School of Science, Jiangsu Ocean University, Lianyungang 222005, China; 2Jiangsu Pacific Quartz Co., Ltd., Lianyungang 222005, China; 3School of Chemistry and Chemical Engineering, Southeast University, Nanjing 211189, China; 4Fujian Key Laboratory of Drug Target Discovery and Structural and Functional Research, School of Pharmacy, Fujian Medical University, Fuzhou 350004, China

**Keywords:** Ru(bpy)_3_^2+^, WS_2_, Au NPs, electrochemiluminescence, carcinoembryonic antigen

## Abstract

In this study, we proposed a novel Ru(bpy)_3_^2+^-Au-WS_2_ nanocomposite (Ru-Au-WS_2_ NCs) nano-hybrid electrochemiluminescence (ECL) probe for the highly sensitive detection of carcinoembryonic antigen (CEA). This system utilizes Au nanoparticles (Au NPs) as a bridge to graft the high-performance of a Ru(bpy)_3_^2+^ ECL emitter and WS_2_ nanosheet with excellent electrochemical performance into an ECL platform, which shows outstanding anodic ECL performance and biosensing platform due to the synergetic effect and biocompatibility of Au NPs and WS_2_ nanosheet. Because the ECL intensity of Ru(bpy)_3_^2+^ is sensitively affected by the antibody-antigen insulator, a preferable linear dependence was obtained in the concentration range of CEA from 1 pg·mL^−1^ to 350 ng·mL^−1^ with high selectivity (LOD of 0.3 pg·mL^−1^, S/N = 3). Moreover, the ECL platform had good reproducibility and stability and exhibited excellent anti-interference performance in the detection process of CEA. We believe that the platform we have developed can expand the opportunities for the detection of additional high specificity-related antibodies/antigens and demonstrate broad prospects for disease diagnosis and biochemical research.

## 1. Introduction

Cancer is one of the leading causes of death in humans, which kills nearly 10 million people worldwide every year. In fact, the previous study found that the early detection and treatment of malignant tumors in early carcinoma and advanced carcinoma would significantly reduce the difficulty coefficient of treatment and mortality. It is well known that carcinoembryonic antigen (CEA) is a tumor marker for different types of cancer such as breast cancer and lung cancer [1]. Therefore, CEA has an important application prospect in the early diagnosis of tumors. Rapid quantitative detection of CEA has important value in clinical diagnosis, treatment evaluation and drug discovery [2]. There are many immunoassays to detect CEA, such as colorimetric, enzyme-linked immunoassay, capillary electrophoresis, fluorescence and electrochemistry [3,4,5,6,7]. Most of these strategies have complicated experimental procedures, unstable substrates and long detection times. Electrochemiluminescence (ECL) as a classical sensing technique, in which the input signal of current and the output signal of light are completely separated, has received considerable attention owing to its high sensitivity [8]. Due to its benefits, such as high selectivity, high sensitivity, rapid response, and low background signal [9,10], the ECL immunoassay has thus developed into a potent analytical tool for the highly specific and sensitive detection of cancer biomarkers such as CEA and prostate-specific antigen (PSA).

Two-dimensional (2D) graphene-like materials such as MoS_2_, WS_2_, MoSe_2_ and WSe_2_ nanosheets, have recently attracted intense interest due to their high dispersibility in aqueous solution, high loading efficiency for biomolecules and good biocompatibility [11,12,13,14]. Through the combination with 0D materials, the graphene-like materials obtain more excellent application prospects. Recent studies have shown that 0D/2D nanocomposites, including MoS_2_-PbS QDs and Au NPs/GQDs-WS_2_ possess potential applications in electrocatalysts, photocatalysis, photodetectors and biosensors [15,16]. For instance, Hou and co-workers fabricated an Au NPs/MoS_2_ heterostructure ECL biosensor for the detection of miRNA-210 in breast cancer tissue [17]. Shorie et al. fabricated the composites of gold nanoparticles on WS_2_ nanosheets by using the intrinsic reducing properties of the nanosheet [18]. Huang et al. [19] combined gold nanoparticles (Au NPs) with WS_2_ nanosheets to construct an electrochemical biosensor for 17β-estradiol. Designing new systems for specialized applications may be encouraged by the synergistic effect of 0D and 2D structures. However, there are few reports on the use of 0D and 2D hybrid nanomaterials for ECL.

As far as we know, 0D and 2D hybrid structure nanomaterials incorporated into an ECL sensor for CEA detection have not been reported. In this paper, we fabricated a hybrid Ru-Au-WS_2_ NCs combining 0D Au NPs, 2D WS_2_ nanosheet and Ru(bpy)_3_Cl_2_ via a cost-effective and conventional sonication approach. Furthermore, an ECL platform was developed for immunosensing CEA by coating Ru-Au-WS_2_ NCs onto a glassy carbon electrode surface and subsequently immobilizing the CEA antibody. Experimental conditions including CEA incubation time and Tripropylamine (TPA) concentration on the behavior of the biosensor were optimized for determining CEA with excellent performances. An ECL immuno-assay method based on Ru-Au-WS_2_ NCs was developed for CEA determination. The hybrid structure with WS_2_ thin-layers and Au NPs improved the electrocatalytic activity of the ECL system. The electrical conductivity, biocompatibility, electrocatalytic activity and large specific surface area of the 0D-Au NPs-2D WS_2_ NS heterostructure were used to detect the actual samples [20]. The constructed hybrid nanomaterials ECL probes displayed simple and fast determination of CEA from 1 pg·mL^−1^ to 350 ng·mL^−1^ with the detection limit down to 0.3 pg·mL^−1^ (S/N = 3) [21].

## 2. Experimental Section

### 2.1. Chemicals and Reagents

CEA and anti-CEA were purchased from Bosai Co., Ltd. (Zhengzhou, China). Bovine serum albumin (BSA) was purchased from Dingguo Co., Ltd. (Beijing, China). TPA, WS_2_, *N*-dimethylformamide (DMF) and Tris(2,2′-bipyridyl) ruthenium (II) chloride hexahydrate were purchased from Sigma-Aldrich (Beijing, China). All other reagents are analytically pure and could be used right away without any additional purification. Each solution was made with double-distilled water.

### 2.2. Instrument and Apparatus

Cyclic voltammetry (CV) and electrochemiluminescence (ECL) measurements were made using an electro-chemiluminescence analyzer, model MPI-B (Xi’an Remex Analysis Instrument Co., Ltd., Xi’an, China). All ECL measurements were performed using a standard three-electrode setup, which included an Ag/AgCl (saturated KCl solution) reference electrode, a platinum wire counter electrode, and a glassy carbon working electrode. The ECL responses of the electrode were recorded in 0.10 M PBS (pH 7.4) containing 100 mM TPA with scanning the potential from 0.2 to 1.4 V at 100 mV/s, and the bias voltage of the photomultiplier tube (PMT) was 800 V. The UV-vis absorption spectra were measured using the UV-vis spectrophotometer Uv-2450 (Shimadzu, Japan). Transmission electron microscopy (TEM) was carried out on a JEOL 2010 TEM with an accelerating voltage of 200 kV.

### 2.3. Preparation of Au NPs

The preparation of Au NPs was conducted as in the previous method [22]. Briefly, 1 mL of 2% HAuCl_4_·4H_2_O was mixed with 79 mL water and then 20 mL containing 0.4% sodium citrate and 0.01% tannic acid solution was quickly added while stirring. The mixed solution was then kept stirring for another 35 min at 60 °C until the mixture turned a deep red color, signifying the formation of Au NPs. Before use, the acquired Au NPs were cooled to room temperature and stored in a brown glass bottle at 5 °C.

### 2.4. Preparation of WS_2_ Nanosheets

To create a uniform, black suspension with a final concentration of 1 mg·mL^−1^, 20 mg of WS_2_ powder with a particle size of 90 nm were dispersed in 20 mL DMF and ultrasonically treated at room temperature for 60 min. After centrifuging the suspension at 12000 rpm for 10 min, the supernatant was discarded and the product was dissolved in 2 mL of water.

### 2.5. Preparation of Ru-Au-WS_2_ NCs

An amount of 150 μL of Ru(bpy)_3_Cl_2_ (50 mM) was quickly mixed with 2 mL of Au colloidal solution. After stirring the reaction mixture for 12 h, 0.5 mL of WS_2_ nanosheet solution was added and continued stirring for 12 h. After being collected, the supernatant was cleaned with distilled water before being centrifuged at 10,000 rpm for 15 min. At long last, the Ru-Au-WS_2_ NCs were scattered in 2 mL of water for assist characterizations and applications. During the whole synthesis process, Ru(bpy)_3_Cl_2_ and Au NPs were first functionalized and mixed, and then Au NPs were grown in situ on the surface of WS_2_ NS, and finally Ru-Au-WS_2_ NCs were successfully synthesized.

### 2.6. Fabrication of the ECL Immunosensor

Alumina powder (1.0, 0.3, and 0.05 μm) was used to polish a glassy carbon electrode (GCE, 3 mm diameter, CH Instruments, Shanghai, China.). The GCE was successively sonicated in 1:1 nitric acid, acetone and distilled water. Following a thorough water wash and immediate N_2_ drying of the cleaned electrode, 5 μL of the Ru-Au-WS_2_ NCs solution was poured onto the GCE surface. After air drying at room temperature, the Ru-Au-WS_2_ NCs modified GCE (GCE/Ru-Au-WS_2_) was produced. After that, a 6 μL coating of a 500 μg·mL^−1^ antibody (Ab) solution was applied to the modified electrode’s surface, which was then allowed to dry at 4 °C overnight. Then, a cross-linking reaction between glutaraldehyde and Ab was used to stabilize the interface. The electrode was then washed with PBS (pH = 7.4) and 3 mL of a 1 wt% BSA solution for 1 h to prevent non-specific binding sites. At this time, the modified electrode was donated as GCE/Ru-Au-WS_2_/Ab/BSA. 

### 2.7. ECL Detection of CEA

For the CEA assay, the prepared ECL immunosensors were put in CEA solutions of different concentrations and allowed to hatch at 37 °C for 1 h. The pH 7.4 PBS was used to carefully wash away any remaining non-chemisorbed CEA before recording the immunosensor’s ECL signals in 0.1 M pH 7.4 PBS containing 0.1 M TPA. Finally, the modified electrode treated with CEA solution was donated as GCE/Ru-Au-WS_2_/Ab/BSA/CEA. Finally, the concentrations of CEA were determined by ECL. The step-by-step preparation diagram of the ECL immunosensor was shown in Figure 1.

## 3. Results and Discussion

### 3.1. Characterization of WS_2_ NS and Ru-Au-WS_2_ NCs 

The WS_2_ and Ru-Au-WS_2_ NCs were characterized by TEM. The layered structure of WS_2_ NS was confirmed by TEM results, as shown in Figure 2a. A large number of Au NPs are uniformly loaded on the surface of WS_2_. Since Ru(bpy)_3_Cl_2_ is positively charged and Au NPs are negatively charged [23], Ru-Au-WS_2_ NCs are aggregated by electrostatic interaction Figure 2b.

To investigate the properties of the synthesized Ru-Au-WS_2_ NCs, the UV-vis absorption spectra of the synthesized Ru-Au-WS_2_ NCs, Au NPs, WS_2_ sheets and pure Ru(bpy)_3_^2+^ were recorded. As shown in Figure 3, the UV-vis absorption peaks at 287 and 454 in the Ru(bpy)_3_^2+^ spectrum were attributed to the π–π* electronic transition of bi-pyridine and metal to ligand charge transfer adsorption, respectively. A typical absorption peak at 526 nm was assigned to the characteristic peak of Au NPs. Furthermore, WS_2_ displays an obvious absorption peak at 633 nm, and three inconspicuous absorption peaks at 530 nm, 460 nm and 420 nm, respectively. Compared to the absorption spectra of Ru(bpy)_3_^2+^ (454 nm), Au NPs (526 nm) and WS_2_ (633 nm), the characteristic absorption peak of Ru-Au-WS_2_ NCs show different degrees of red-shift. All the above results demonstrate the successful preparation of Ru-Au-WS_2_ NCs.

### 3.2. Electrochemical and Electrochemiluminescence Characterization of Modified Electrode

CV was used to electrochemically describe the immunosensor’s assembly process. At a scan rate of 100 mV s^−1^ in a solution of 5.0 mmol·L^−1^ potassium ferricyanide, all of the cyclic voltammetry curves were obtained within the potential range of −0.2–0.4 V, as shown in Figure 4. The bare electrode has an anodic peak at about 0.3 V, and a cathodic peak at 0.2 V, and the peak current is the largest compared to the other curves (curve a). After the Ru-Au-WS_2_ NCs were assembled, the peak current decreased slightly (curve b). Furthermore, the Ru-Au-WS_2_ NCs provide a large number of binding sites for anti-CEA immobilization. When 100 ng·mL^−1^ anti-CEA was added to the electrode surface, the peak current continued to fall, proving that the antibody had been successfully immobilized on the surface (curve c). The peak position difference and the peak current both significantly increased when BSA was used to block unbound antibody sites on the electrode surface (curve d). Additionally, the peak current decreased significantly due to the varying degrees of electron obstruction between anti-CEA and BSA. When 10 ng·mL^−1^ CEA antigen was added to the electrode, the peak current was further reduced (curve e). The specific antigen-antibody binding between anti-CEA and CEA, and the generated immune complex hindered the rate of electron transfer in the composite membrane, resulting in a decrease in the response current, which may account for this result.

In this experiment, TPA was selected as the co-reactant. A three-electrode system was used, with the working electrode being GCE/Ru-Au-WS_2_, the reference electrode being Ag/AgCl (saturated KCl), and the auxiliary electrode being Pt wire. The ECL properties of Ru-Au-WS_2_ NCs film were investigated in 0.1 mol·L^−1^ TPA solution using the reference experimental conditions (pH 7.4, PBS, 37 °C incubation for 60 min). The results are shown in Figure 5. As shown in (curve a), the synthesized Ru-Au-WS_2_ NCs has a strong ECL signal under experimental conditions, which also indicates that the Ru-Au-WS_2_ NCs were successfully prepared. As shown in (curve b), when the Ru-Au-WS_2_ NCs modified electrode was immobilized by Ab and a cross-linking reaction between glutaraldehyde, the ECL signal was significantly reduced, demonstrating that the antibody had been successfully immobilized on the electrode surface. Immediately afterward, as shown in (curve c), the unbound antibody site on the electrode surface was blocked with 0.5% BSA solution, and the ECL signal was found to decrease again. Finally, when the electrode was modified with 10 pg·mL^−1^ CEA antigen, the ECL signal kept declining, as shown in (curve d). This result should be attributed to the specific antigen-antibody binding between anti-CEA and CEA, and the generated immune complex once again hindered the transfer of electrons in the composite films. Consequently, the quantity of [Ru(bpy)_3_^2+^]* produced by Ru(bpy)_3_^2+^ reaction was decreased, which decreased the ECL signal.

The effects of CEA incubation time and TPA concentration on the behavior of the biosensor were examined to maximize the performance of the manufactured immunosensor. The best signal from the biosensor platform was produced when the TPA concentration was 0.1 mol·L^−1^ and the culture time was 30 min, as shown in Figure 6A,B. As a result, these circumstances were chosen as the best parameters for further CEA detection.

### 3.3. Quantitative Detection of CEA by Immunosensor 

In addition, different concentrations of CEA antigen were incubated under the condition of GCE/Ru-Au-WS_2_/Ab/BSA modified electrodes as substrates. The quantitative detection of CEA antigen was realized in 0.1 mol·L^−1^ TPA solution. As shown in Figure 7, the ECL signal exhibits a decreasing trend as CEA antigen concentration rises. This can be explained as follows: with the increase in CEA concentration, anti-CEA has a specific antigen-antibody binding effect with CEA, and the immune complexes generated at the immune sensing interface increase, which continuously hinders the transfer of electrons in the composite film, so that the quantity of [Ru(bpy)_3_^2+^]* produced by Ru(bpy)_3_^2+^ reaction gradually declines, the ECL signal gradually declines. As a supplement, the linear range of the response of the sensor used in the experiment to CEA of different concentrations is 1 pg·mL^−1^–350 ng·mL^−1^, the linear equation is *i*_p_(*μ*A) = −707.06178 + 2547.16748 logC_(CEA)_ (ng·mL^−1^), and the detection limit is 0.3 pg·mL^−1^. 

The selectivity of the designed biosensor was studied shown in Figure 8A. HSA (20 ng·mL^−1^), prostate-specific antigen (20 ng·mL^−1^), immunoglobulin G (20 ng·mL^−1^), and alpha-fetoprotein (20 ng·mL^−1^) interferences were added to 10 ng·mL^−1^ CEA at two times the concentration. Comparing the ECL reaction allowed for the evaluation of the specificity. The ECL signal of the mixture containing CEA alone (10 ng·mL^−1^) and HSA, PSA, IgG, and AFP was not significantly different from each other. The stability of the designed biosensor studied was shown in Figure 8B; after successive scans for 12 cycles, the ECL intensity of GCE/Ru-Au-WS_2_/Ab/BSA/CEA almost keeps constant, which manifests the stability of the designed biosensor.

Table 1 demonstrated some characteristic comparisons between Ru-Au-WS_2_ ECL biosensor and other 2D graphene-like materials methods. 

## 4. Conclusions

In conclusion, an ECL immunoassay assembly is fabricated using the 0D Au NPs and 2D WS_2_. The application of graphene-like WS_2_ material effectively increased the loading of Ru(bpy)_3_^2+^ on the surface of the electrode and exhibited high ECL intensity. Additionally, Au NP amplification was used to increase the detection sensitivity for protein ECL analysis. High sensitivity, a broad linear range, good reproducibility, and acceptable precision and accuracy were all displayed by the proposed immunosensor. This work offers a fresh immobile substrate for the identification of additional tumor markers. Designing new ECL systems for biosensor applications may be aided by the synergy between the 0D and 2D structures.

## Figures and Tables

**Figure 1 biosensors-13-00058-f001:**
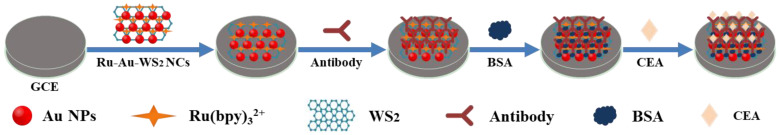
The schematic preparation process of immunosensor.

**Figure 2 biosensors-13-00058-f002:**
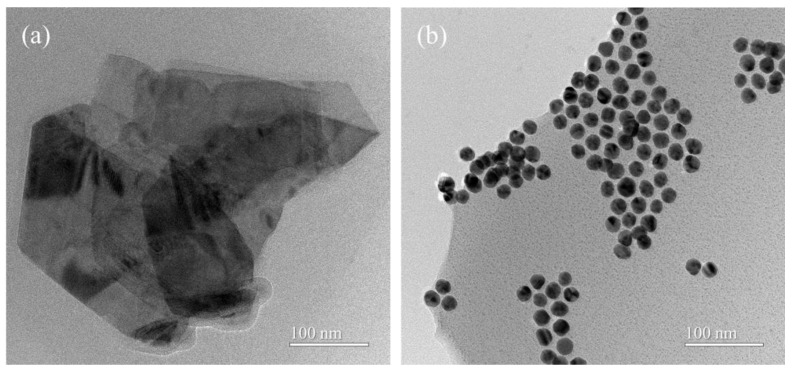
TEM images of (**a**) WS_2_, and (**b**) Ru-Au-WS_2_ NCs.

**Figure 3 biosensors-13-00058-f003:**
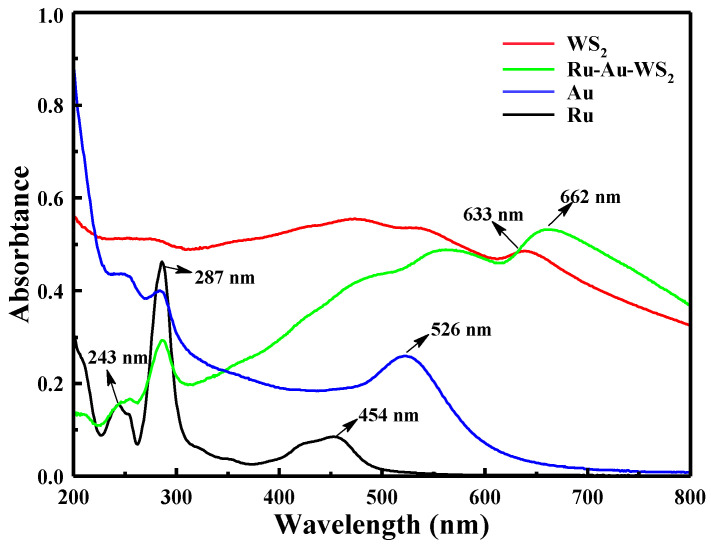
UV-vis absorption spectra of Au, Ru, WS_2_ and Ru-Au-WS_2_ NCs.

**Figure 4 biosensors-13-00058-f004:**
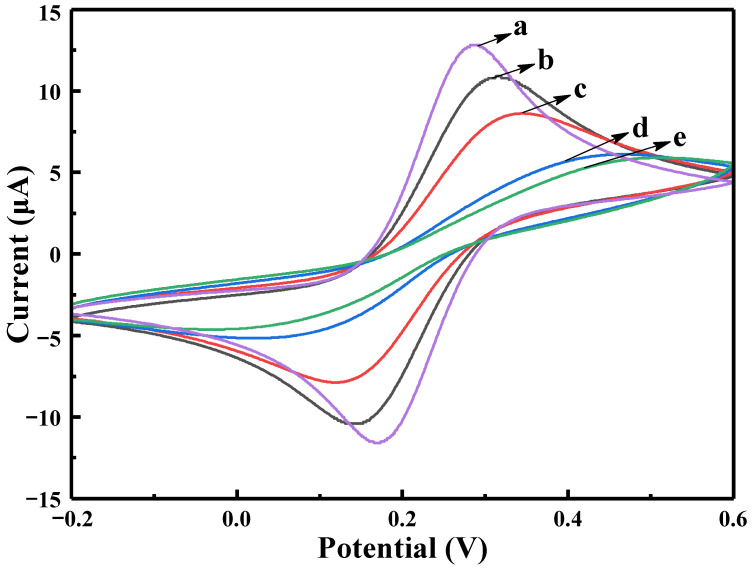
Cyclic voltammetric test diagram of modified electrodes in different states: (**a**) GCE, (**b**) GCE/Ru-AuWS_2_, (**c**) GCE/Ru-Au-WS_2_/Ab, (**d**) GCE/Ru-Au-WS_2_/Ab/BSA, (**e**) GCE/Ru-Au-WS_2_/Ab/BSA/CEA.

**Figure 5 biosensors-13-00058-f005:**
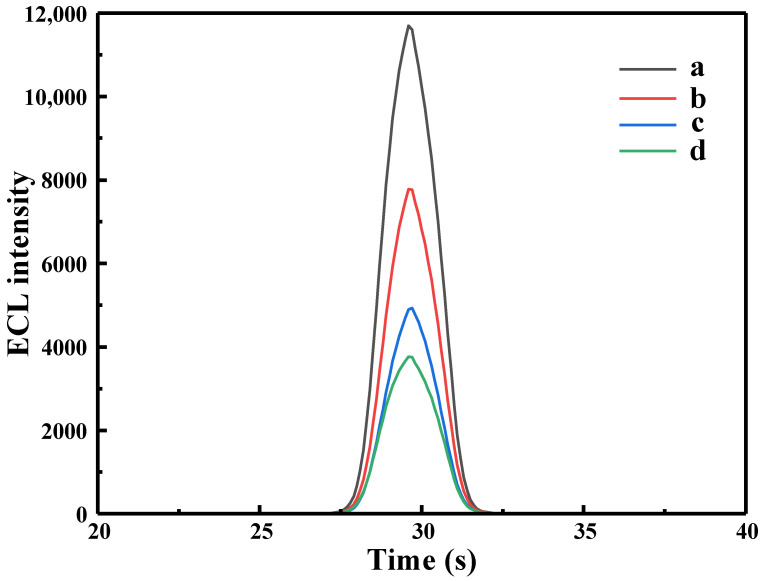
ECL diagram of modified electrodes in different states: (**a**) GCE/Ru-Au-WS_2_, (**b**) GCE/Ru-Au-WS_2_/Ab, (**c**) GCE/Ru-Au-WS_2_/Ab/BSA, (**d**) GCE/Ru-Au-WS_2_/Ab/BSA/CEA.

**Figure 6 biosensors-13-00058-f006:**
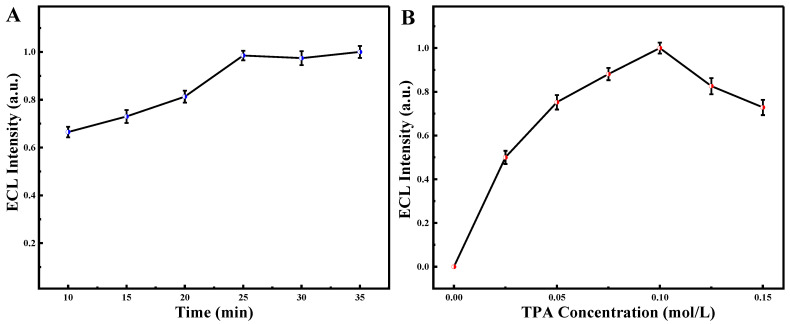
The effects of incubation time of CEA (**A**) and the concentration of TPA (**B**).

**Figure 7 biosensors-13-00058-f007:**
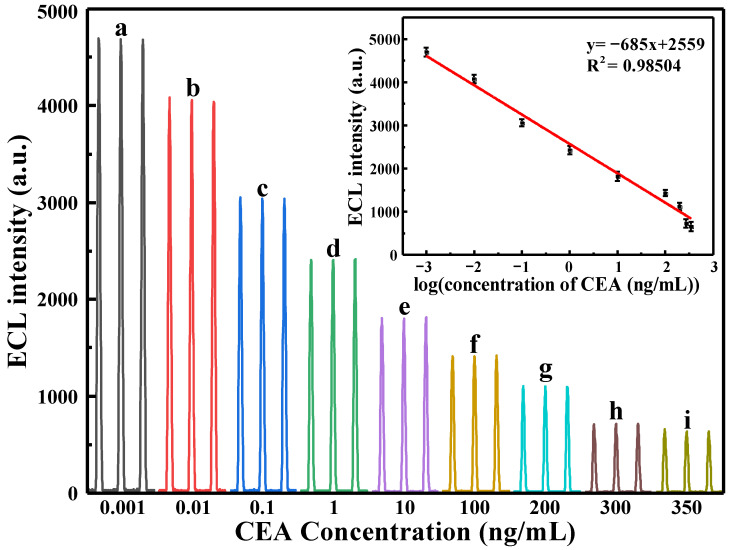
The ECL curve for quantitative detection using GCE/Ru-Au-WS_2_/Ab/BSA, the antigen concentration from left to right is: (**a**) 0.001, (**b**) 0.01, (**c**) 0.1, (**d**) 1, (**e**) 10, (**f**) 100, (**g**) 200, (**h**) 300, (**i**) 350 ng·mL^−1^, the inset shows the linear relationship between ECL signal and antigen concentration.

**Figure 8 biosensors-13-00058-f008:**
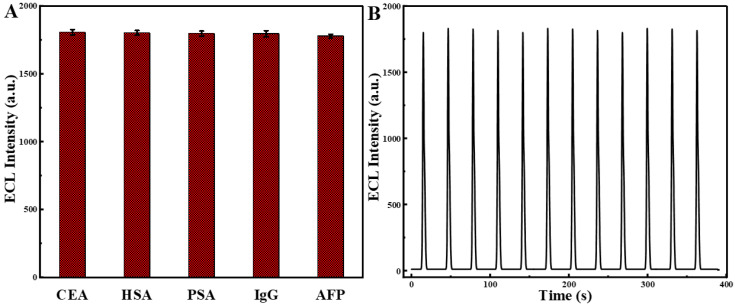
The selectivity of CEA 10 ng·mL^−1^ and CEA 10 ng·mL^−1^ mix 20 ng·mL^−1^ Human Serum Albumin (HSA), prostate-specific antigen (PSA), immunoglobulin G (IgG) and alpha-fetoprotein (AFP) (**A**) and the stability of GCE/Ru-Au-WS_2_/Ab/BSA/CEA (**B**).

**Table 1 biosensors-13-00058-t001:** A comparison of different methods for the detection of CEA.

Immunosensor	Detection Method	Linear Range (ng·mL^−1^)	LOD (pg·mL^−1^)	Refs.
Anti-CEA(BSA)/Au/WS_2_/GCE	PEC	0.001–40 ng·mL^−1^	0.5 pg·mL^−1^	[24]
AFW-CA72-4-Ab2/CA72-4/CA72-4-Ab1/IBM/GCE	DPV/CV	2–50 U/L	0.6 U/L	[25]
anti-CEA/Fe_3_O_4_ NPs@Au NPs/d-Ti_3_C_2_T_X_ MXene	SERS	0.0001–100 ng·mL^−1^	0.033 pg·mL^−1^	[26]
MoS_2_/g-C_3_N_4_-PtCu/Ab_2_/BSA/CEA/Ab_1_/MoS_2_-Au NPs/GCE	*i-t*	0.0001–80 ng·mL^−1^	0.03 pg·mL^−1^	[27]
nanoprobe/CEA/BSA/anti-CEA/MoS_2_-AuNPs/GCE	DPV/CV	0.00001–1 ng·mL^−1^	0.0012 pg·mL^−1^	[28]
GCE/Ru-Au-WS_2_/Ab/BSA/CEA	ECL	0.001–350 ng·mL^−1^	0.3 pg·mL^−1^	This work

## Data Availability

Not applicable.

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
