# Peer review of "Ultrasensitive Electrochemiluminescence Immunoassay Based on Signal Amplification of 0D Au—2D WS2 Nano-Hybrid Materials"

_biosensors, 2022, doi:10.3390/bios13010058_

Round 1

Reviewer 1 Report

The authors present the fabrication and testing of an ECL-based immunoassay for CEA. In addition to the comments in the attached PDF, my suggested revisions and inclusions for the manuscript are as follows:

·         The introduction lacks detail about the mechanism of ECL used in this system and needs to be added to give the reader a clear understanding of how the assay operates.

·         The paragraph (lines 138-151) needs to be more precise, as the information is jumbled. Also, ref 24 does not make mention of the origin of the transitions.

·         Line 185, there is no proof of covalent bonding, and it is also unclear what the authors are referring to. Is it the NC that is supposedly covalently bound, or is it the antibody? Also, where does the antibody bind? Is it to the electrode directly or via the NC?

·         Figure 6- Are they error bars? If so, is it standard error or standard deviation that they refer to. Also, how many replicates were conducted?

·         Figure 7- Why is the x-axis time when the signals differ according to concentration? Also, at least 3 replicates should be done for each concentration for statistical relevance. What is the r-value for the linear relationship.

·         The pulse sequence for the ECL analysis needs to be included.

Reviewer 2 Report

This work reported a facile electrochemiluminescence immunoassay platform based on ECL probe Ru(bpy)32+-Au-WS2 nanocomposites (Ru-Au-WS2 NCs). The Ru-Au-WS2 NCs were prepared by utilizing the strong coordination effect of carboxyl groups of Au NPs with tungsten atoms of WS2 nanosheets. This strategy is interesting and might be a good guidance for the construction and applications of ECL sensor. Meanwhile, the paper is well prepared and data are quite reasonable to demonstrate the points. I recommend its publication after minor revision.

1. Please describe in detail the original intention of doing this work. In other words, what’s most important innovation in this work?

2. Results and discussion section: the synthesis mechanism of the Ru-Au-WS2 NCs should be explained in detail.

3. The error bar should be given in the inset of Figure 7: the linear relationship between ECL signal and antigen concentration.

4. This paper is innovative in ECL sensor strategy, please supplement the prospect of the later research.

5. A few format mistakes should be corrected, such as:

(1) The full name should be given the first time the abbreviation TPA appears.

(2) Journal names in reference should be Abbreviated.

(3) Page 3, line 132, WS2 should be changed to “WS2”.

(4) In the abstract section: the full name of ECL should be supplemented when the abbreviation first appeared.
